# A social engineering model for poverty alleviation

Amit K. Chattopadhyay [1✉], T. Krishna Kumar[2] & Iain Rice[3]

Poverty, the quintessential denominator of a developing nation, has been traditionally defined against an arbitrary poverty line; individuals (or countries) below this line are deemed poor and those above it, not so! This has two pitfalls. First, absolute reliance on a single poverty line, based on basic food consumption, and not on total consumption distribution, is only a partial poverty index at best. Second, a single expense descriptor is an exogenous quantity that does not evolve from income-expenditure statistics. Using extensive income-expenditure statistics from India, here we show how a self-consistent endogenous poverty line can be derived from an agent-based stochastic model of market exchange, combining all expenditure modes (basic food, other food and non-food), whose parameters are probabilistically estimated using advanced Machine Learning tools. Our mathematical study establishes a consumption based poverty measure that combines labor, commodity, and asset market outcomes, delivering an excellent tool for economic policy formulation.

[1] Aston University, Department of Mathematics, Birmingham B4 7ET, UK. [2] Rockville-Analytics, Rockville, MD 20850, USA. [3] Birmingham City University, Millennium Point, Birmingham B4 7XG, UK. ✉email: a.k.chattoadhyay@aston.ac.uk

P overty is associated with the living standards of low income households. There are three ways of looking at poverty. In the first approach, poverty is defined as the deficiency in the level of living, measured through insufficient consumption of the essential commodities for low-income persons, who spend all their income on essential commodities. This was the approach used by Engel[1–3].

The second and more commonly used poverty approach depends on a poverty line that is obtained independently and exogenously. This trend started in 1901 with Rowntree[4], who defined poor as individuals with income below the poverty line level of income needed to cover basic needs. Popular approaches, such as those in refs. [5,6], used consumption data to determine an arbitrary poverty line only, thus failing to measure consumption deprivation (CD). Arbitrariness is thus inherently embedded in the definition of the poverty line. While the U.S. poverty line is measured at three times the money needed to buy a low-income diet plan outlined by the US Department of Agriculture[7], the Indian poverty line is determined by assuming a particular percentage of urban poor[8]. This approach, based on a subjectively determined poverty line, is not helpful in bringing objectivity to the poverty studies, for a scientific theory of poverty. Sen developed an axiomatic theory of poverty index[9]. His Focus axiom axiomatized Rowntree's notion of an income poverty line, below which people are labeled poor. As axioms are often taken for granted and rarely questioned, poverty line had remained the quintessential benchmark in the field for long, despite obvious controversies surrounding its origin. Foster et al. came up with a class of poverty indices dependent on a poverty line[10], such that all other known poverty indices arose as special cases.

There is the third approach, now an accepted wisdom of multidimensional poverty[11–13]. This existing literature on multidimensional poverty also requires multiple subjectively determined thresholds (poverty lines). All these three approaches suffer from two drawbacks that they do not link poverty or CD to the general working of the broader economic system that generates this poverty, and they do not provide a method of predicting the future level of poverty as a function of controllable economic parameters.

We draw upon our earlier work in integrating all these elements. Fox and Kumar argued that there is a spatial hierarchy of markets under the assumption of lexicographic preferences and separable utility functions[14]. Sitaramam et al. showed that while the Engel curve for the most essential commodity is concave with respect to income, the next most essential commodity is also concave with respect to the residual income, and so on. They[15] established empirically that there exists a commodity hierarchy of needs, complementing a series of concave Engel curves as functions of residual income, leading to a phenomenological representation of the Engel curve[16,17], using a Michaelis–Menten model[18], where the concave function represented real consumption of cereals (which ranked as the first basic need) as a function of income, proxied by total expenditure. They used the cereal CD as the deprivation function of Atkinson[19] to demonstrate that cereal CD index serves as a poverty index. Kumar et al. later used a much larger data base (made available in digitized form by the National Sample Survey Organization) at individual level from three much larger quinquennial surveys[20], and revalidated the model for all of India and for rural and urban India and for all the Indian states. They also established that the results vindicated the development policies followed by India with respect to growth and growth with equity.

Structured on stochastic wealth exchange between agents close to the poverty line of income, Chattopadhyay et al. developed a model of market exchange to generate income distribution[21] abiding Sen's axioms[22]. Chattopadhyay et al.[23] then applied this

stochastic portfolio on labor markets, commodity markets, and asset markets to link the poverty index with the entire working of the economy, which is so essential for economic policy to reduce poverty[24]. As a consequence, hierarchical Engel curves could be seen to saturate with the rise in residual income, eventually converging to a point where some residual income is left after meeting all essential hierarchical needs. It is that threshold, identifying economic inflexion, that is relevant to study the living standards as conceived by Engel. These three studies[21–23] jointly established a theoretical basis for an endogenous poverty threshold for its comparison over space and time. The time varying income and consumption series from such a model enable us to derive a probabilistically validated poverty index that could predict future poverty lines.

These studies, however, had one fundamental limitation. They used basic food as the lone denominator of poverty, not the well-recognized multidimensional construct involving all possible expenditure modes[11–13,25]. Multidimensional Poverty Indices (MPIs) are mainly of two types. The first category of studies defines the poor as those who are poor at least for one dimension[13,25] while the other category identifies one as poor only when poverty spans all dimensions (acute multidimensional poverty)[12]. But they suffer from the same limitations as the income poverty measures, as they have arbitrary thresholds for minimum needs for each of the chosen dimensions. Furthermore, they combine different dimensions using arbitrary weights that are not data conforming.

For a holistic poverty index, one needs to assimilate information from all three consumption entries and their covariances. In order to suitably weigh all three components of consumption and thereby appropriately address the multivariate problem, here we use advanced machine learning techniques to dimensionally reduce a multi-variable description to a single highest weighted one-dimensional Principal Component (PC) that combines information from all three expense modes. This statistically reduced variable is then modeled within the Fokker–Planck framework as in refs. [21,23] to analyze the time evolution of the probability density of expenditure, relating to the poverty threshold. The recursive details of CD, involving all expense modes, thus leads to a multidimensional, commodity exchange-based poverty index that supersedes previous evaluations and lays the groundwork for a generalized poverty index.

## Results

**Dimensionally reduced Engel curve**. Engel[2] empirically derived a law showing that while expenditure on food consumption initially increases with income, soon enough, such expenses reach a plateau and are hence substituted by other expenditures (like other food, non-food, etc.). Mathematically, this is represented by a concave function (Fig. 1), typically of the form $V(y) = Vy/(K + y)$[15,16]. The real situation is more complicated though due to the presence of multiple expenditure modes—on cereals, other-food, and non-food.

To analyze this, we used NSS income data ($y$)[26] and correlated that with expenses $C_1(y, t)$, $C_2(y, t)$, and $C_3(y, t)$, the latter respectively representing expenses on cereals, other-food, and non-food items. To address this multivariate data modeling, we used independent multivariate algorithms—Neuroscale[27–29], Locally Linear Embedding (LLE)[30], Isomap[31], Curvilinear Component Analysis (CCA)[32], Principal Component Analysis (PCA)[33]—thereby combining all three expense heads into a single (non-dimensional) function to obtain a multivariate version[34] of the Engel plot (Fig. 1).

As can be seen from Fig. 1, which represents the economic equivalent of the Michaelis–Menten law in biochemistry, barring

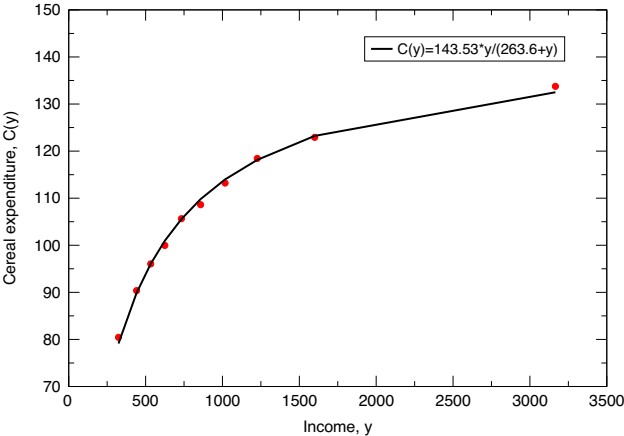

**Fig. 1 Plot of expenditure C(y) against total rescaled income y for 2004 Indian data[26].** The dots represent actual data while the solid line is extrapolation using the formula $C(y) = Vy/(K + y)$, where $V = 143.53$, $K = 263.6$. This Engel curve fits perfectly with the data and validates the machine learned stochastic model.

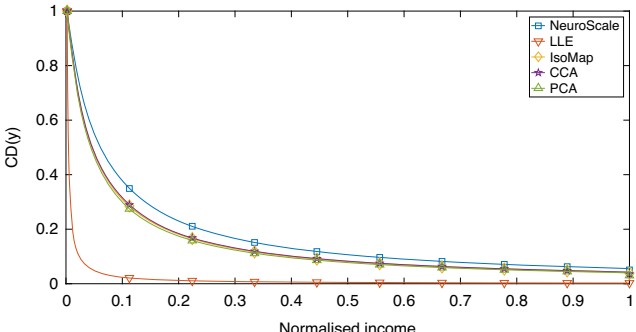

**Fig. 2 Variation of the consumption deprivation function against income.** Plots of $CD(y)$ for the dimension reduced models. As expected, the deprivation drops as income increases up to a saturation point.

fluctuations in the values of fitting parameters $V$ and $K$[21,23], all algorithms conform to the functional fit

$$C(y, t) = \frac{V(t)y}{K(t) + y}, \qquad (1)$$

where the time-dependent parameters $V(t)$ and $K(t)$, respectively, enumerate the multivariate form of the maximum expenditure per family over all three expense heads, and the income required to support half of the maximum multivariate expenditure[20,21,23].

In Fig. 2, the LLE dimension reduced model indicates a far higher CD as would be expected with focus on non-food expenditure. In order to perform this analysis robustly accounting for the three dimensions of our data, we proceed with the NeuroScale dimension reduction model[26]. The IsoMap, CCA, and PCA embeddings show a less significant deprivation here, but the NeuroScale embedding shows a lower level of decay in CD than the other dimension reduced models.

As explained in ref. [15], CD relates to the actual shortfall in the total expenditure from the maximum possible lifestyle based on total consumption alone and is defined as

$$CD(y) = V(t) - C(y) = \frac{V(t)K(t)}{K(t) + y(t)}. \qquad (2)$$

The scalar function $CD(y)$ above is a dimensionally reduced projection of the three-dimensional vector space constituted of cereal, other-food, and non-food expenditure modes. Our

description henceforth will always tacitly assume $C(y(t))$ as the dimensionally reduced scalar projection from the multivariate distribution. Figure 2 plots the CD functions arising out of the respective algorithms (scalar projections, as normal).

As in ref. [23], our target will be to associate a probabilistic structure to the time varying Indian consumption and total expenditure (income) data, the novelty here being the multivariate aspect involving all three modes of expenditure. In order to perform this analysis robustly accounting for the three dimensions of our data, we proceed with the NeuroScale dimension reduction model[27]. The non-constant weights of the three components of expenditure seem to mimic the prices of those three components, as the principal component is almost like a 45° line when plotted against total expenditure. This could be verified against price data derivable from NSSO[26], that will confirm our agent-based model's veracity in predicting equilibrium quantities and prices from consumer expenditure data (details in the online-appendix).

**Machine learning of multivariate statistics.** Figure 3a, b below explains the implication of relative multivariate weighting that some of the more popular dimensional reduction mechanisms will do. Subjective to the (Indian) data analyzed, we find that predictions from all five measures—Neuroscale[27–29], LLE[30], Isomap[31], CCA[32], PCA[33]—converge to reasonable agreement amongst themselves.

While the weights used in the traditional MPI are arbitrary, the weights we use are derived through dimension reduction algorithm. In order to assess the significance of each of the observed expenditure streams (cereals, other food, and non-food), we present Fig. 3b. It may be observed that the three projections add to the 45° line, suggesting that the scalar weights are actually the market prices. This observation is also independently supported by the observations made later about how the relative weights move as income increases.

Figure 3 shows the comparison of the dimension reduced plots with respect to the individual PDFs. However, we can also assess the effectiveness of the dimension reduction methods as a descriptor for the total income realized by each of the expenditure streams; cereals, other food, and non food.

In the PCA dimension reduction case of Fig. 4a, the weights allocated to cereal, other-food, and non-food variables are 0.1694, 0.4336, and 0.8851, respectively. These weights seem to be in the same order of relative magnitude as the market prices for the three components. It is likely that these weights correspond to market equilibrium prices for the three components. It is thus very reassuring that the agent-based model (ABM) is capable of generating equilibrium prices and quantities, and thus fully characterize the competitive market mechanism.

In Fig. 4b, we show the plot of the normalized dimension reduction models against total expenditures (the normalized summation of cereal, other food, and non-food expenditures) for a sample year. These dimension reduced curves are contrasted with a unit curve whereby the dimension reduction model would perfectly account for total expenditure. As shown in Fig. 3a and b, the preservation is highest for NeuroScale and Isomap and lower for the local methods of LLE and CCA as well as for the linear model PCA. Figure 4b thus demonstrates that despite the reduction of dimensions from three in the observation space to a univariate latent model, the information governing the observations can still be preserved.

In order to visualize the relative importance of each dimension in the mapping, we follow the approach of refs. [27,28] to generate three independent batches. In the first batch, we take 100 linear steps between the minimum and maximum values of cereal

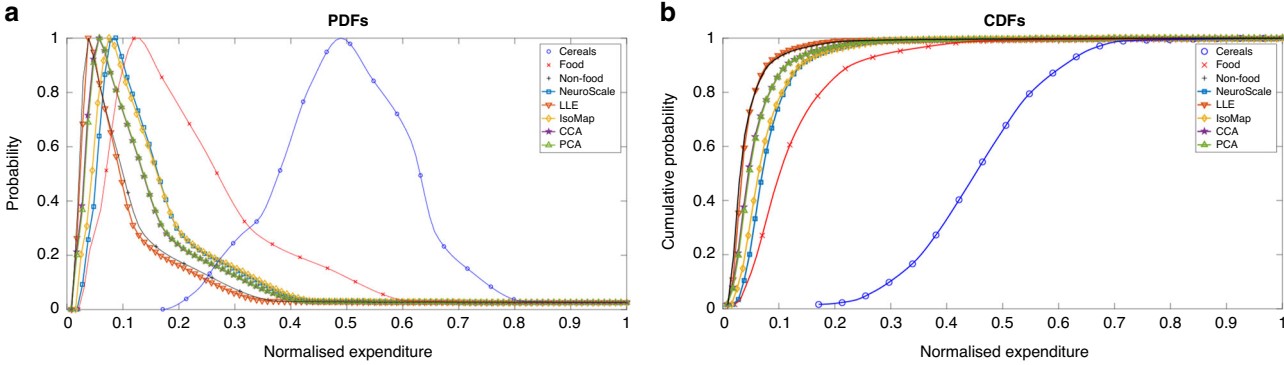

**Fig. 3 Probability Density Function (PDF) and Cumulative Distribution Function (CDF). a** Probability density functions (PDFs) and **b** cumulative distribution functions (CDFs), data modeling over 30 years' of NSS data (1959–1991)[26]. The expenditure on the three categories of cereals (blue circles), other food (red crosses), and non-food (black +'s) is shown for comparison. The local methods such as LLE, PCA, and step-CCA focus significantly on the non-food category, whereas the global mappings of IsoMap and NeuroScale have highest density between the non-food and other food categories. These methods have accounted for the spread of data in more than the single dominant dimension of non-food allowing for a more robust low-dimensional embedding of the data.

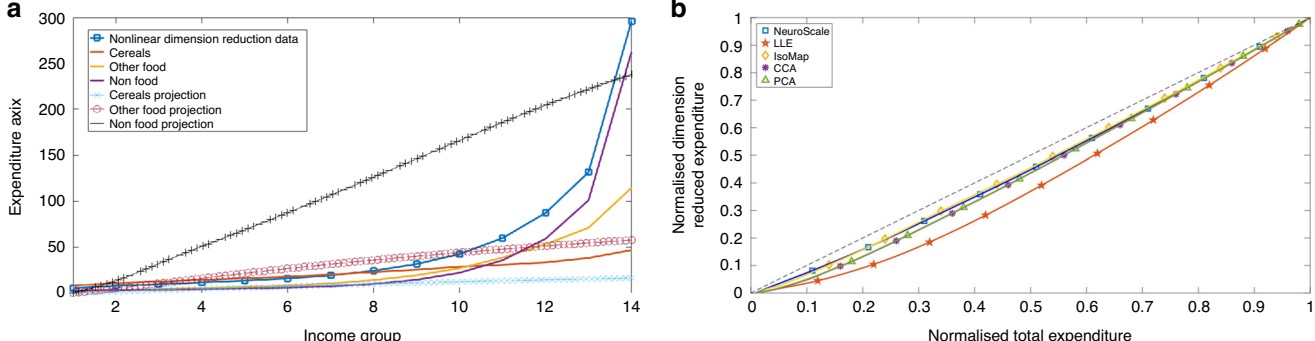

**Fig. 4 Expenditure versus Income - Individual Expense Category against Multivariate Distribution.** Expenditure versus income plots (**a**) and machine learned multivariate projections (**b**). The y-axis variables are $(y, z_{NS})$, $(y, z_{lle})$,.. and $(y, z_{PCA})$, where $z_{NS}$ is the NeuroScale projection of the data to the univariate model etc., and $z_{PCA} = w1 \times C1 + w2 \times C2 + w3 \times C3$, where $\sum_{i=1}^{3} w_i = 1$, $C_i$ ($i = 1, 2, 3$) being the three eigenvectors and $w_i$ are the normalized machine learned eigenvalues.

expenditure keeping the other two dimensions fixed at the median values and project this through the NeuroScale mapping to achieve the univariate curve $z_1$. Repeating this process for the other-food and non-food streams results in the curves $z_2$ and $z_3$. These three variable weights demonstrate relative impact of these three dimensions in the total expenditure/income space. The relative significance of these is shown in Fig. 5, from the curves of $z_2/z_1$ and $z_3/z_1$, both normalized with respect to the equivalent cereal expenditure $z_1$. Individually, they represent the relative projection weights of other-food and non-food, relative to the projection weights for cereals.

Our dimension reduction arises as nonlinear projection of each dimension on the total expenditure. The weights therefore are not constant, as in most MPI studies, but vary with income. At low expenditure levels, other-food has greater weight than non-food in our index (both of them have larger weight than basic food, as the market values them that way). As expenditure increases, both other-food and non-food weights increase. But the weight of other-food remains stable and maintains a level of ten times that of food weight. On the other hand, the weight of non-food keeps on increasing. This demonstrates that people keep on substituting more and more of high priced non-food items as income increases.

Cereal has the lowest weight, other food and non-food have higher weights than cereal. These seem to reflect the order of magnitude of the price indices for the three dimensions. In fact, as

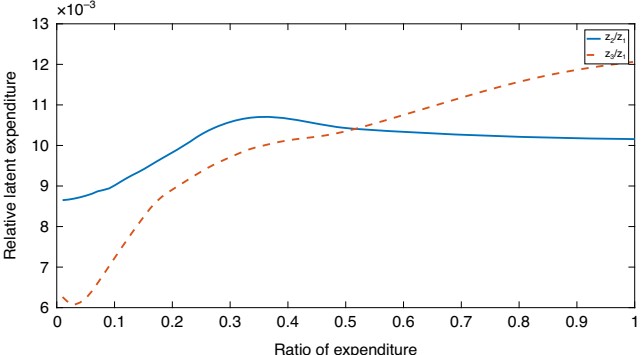

**Fig. 5 Expenditure modes compared.** Relative normalized expenditure under individual heads plotted against the relative total expenditure (the normalized summation of cereal, other food, and non food expenditures); $z_1$ represents the projection weights for cereal expenditure while $z_2$ and $z_3$, respectively, represent projections weights for other-food and non-food-based expenditures.

our data are generated from the outcomes of market transactions, it is natural to expect that the projection would only discover the patterns in market exchange, and the weights mimic the market prices. Thorbecke[11] describes how the market prices could arise as weights by using the indirect utility functions. He also shows

the normative limitations associated with market-based weights, as they do not correct for market distortions and do not capture non-market transactions. The implication of this observation is that when the markets fail to address the needs of the deprived, necessary corrective steps have to be taken to alter the allocation mechanism. Another issue discussed is the inherent dependence of the various dimensions. While the details are beyond the scope of the present study, it must be noted that the dimension reduction algorithm uses a distance metric, e.g., Mahalanobis' distance metric[3] accounting for the correlations between different dimensions. The observed non-linear weighting pattern reflects the fact that the prices vary with incomes, reflecting the variation in the quality of the goods and quality of the markets through which the goods are exchanged (Fig. 5).

**The multivariate poverty index.** The definition of the poverty index $P_{CD}(t)$ follows the structure laid out in refs. [16,19,21,23]. The key difference is the substitution of the cereal expenditure variable $C(y)$, as in refs. [21,23], with the all expense mode inclusive multivariate-$C(y)$ obtained by dimensional reduction, as detailed above.

$$P_{CD}(t) = \int_{y_0}^{\infty} CD(y,t)\hat{f}(y,t)\,dy, \qquad (3)$$

where $CD(y,t)$ and $\hat{f}(y,t)$ will be, respectively, obtained from Eqs. (14) and (12). The time varying mean income $C(t)$ present in Eq. (14) can be evaluated from the relation $C(t) = \int_{y_0}^{\infty} yf(y,t)\,dy$ which addresses the oscillatory instability that would otherwise destabilize the numerical simulation should the $C(t) = 1.16 \times t - 2218.7$ fit function be used instead (details in ref. [23]). $V(t)$ and $K(t)$ used in Eq. (14) above are extrapolation fits of data, as shown in Fig. 6. The strength of our complementary model can be justified both from the fit (solid line against real data circles) as well as from the solution (dotted line) of our proposed model represented in Eq. (14).

As shown in Fig. 6, linear regression fits over 23 years' NSS data[26] suggest poverty increasing from 1951 to 1970 and then decreasing steadily. The peak is shown immediately in the pre-1970 era. India had war with China in 1962. It had a war with Pakistan in 1965–66, it had a draught in 1965. USA stopped its aid to India in 1965. The war resulted in price rise, including for food grains. All these factors were responsible for the rise of poverty in India prior to 1969. The green revolution also started working from 1970. India also launched poverty alleviation programs in the Fifth Five Year Plan (1974–79). These developments explain the sharp decline in poverty between

1970 and 1990. India's economic liberalization through more market-friendly economic policies that started in 1991 arrested the sharp declining trend in poverty and slightly flattened it after 1991.

Another key aspect of the multivariate index is its relatively high measure compared to the HCI and PG indices, as also noted in other multidimensional studies related to the human development initiative[12,13]. A data based reconnaissance affirms that according to the global MPI statistics recorded in 2018, 271 million Indians moved out of poverty between 2005–06 and 2015–16, https://ophi.org.uk/multidimensional-poverty-index/global-mpi-2018/, which effectively implies that the Indian poverty rate has nearly halved, an estimate that is not shown accurately in the unidimensional measures. Consideration of all modes of expenditure essentially contributes in further reducing the magnitude of the poverty index. This is most noticeable in multidimensional PI measures[12,13] where the reduction in the overall PI with addition of the non-food and other-food-related extra dimensions. This is nothing special of the Indian data that we used here but is seen elsewhere as well[35]. As a confirmation, we provide multidimensional analysis of the US data in the online Supplementary Information to affirm this statement.

We compared the new theoretical poverty index in Fig. 7 with data from all three indices popularly used in the literature: the head count (HCI) index, the poverty gap (PG) index, and the squared poverty gap (SPG) index. Our most recent work[23] showed two key incongruences with its earlier data-based predecessor[21], notably the dips in the poverty index (versus time) plot at years 1970 (sharper in ref. [21] than in ref. [23]) and then at 1987 (simply not showing up in ref. [23]) as well as in the present work. The first is now reckoned to be an artifact of the nature of NSS data in that in the pre-1970 regime, economic data were not appropriately calibrated against corresponding expenditure heads. This deficiency shows up in our present model as well but one must remember that a statistical predictive model is unlikely to forecast local dips crests and troughs. Its strength is in its ability to reach beyond data by incorporating all expense heads and then crystalizing such information within the realms of a dimensionally reduced, effectively single variable model.

**Discussion.** The importance of multidimensional poverty assessment is now well acknowledged in the literature[12,13], especially its preponderance in Indian poverty estimation[35]. The critical issue though, lay in the mechanism of estimation that

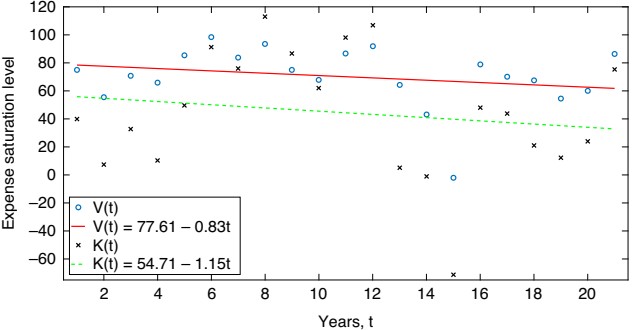

**Fig. 6 Engel parameter dynamics.** $V(t)$ (solid line) and $K(t)$ (dotted line) of the dimensionally reduced model versus time $t$. As shown in the figure, the regression fit functions are given by $V(t) = 77.61 - 0.83t$ and $K(t) = 54.71 - 1.15t$. The regression fits justify the first-order linearity approximations of $V(t)$ and $K(t)$ as functions of time $t$.

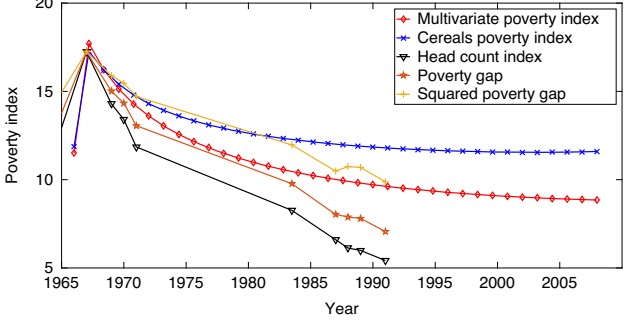

**Fig. 7 Plots of multivariate poverty indices for the dimension reduced model.** As expected, the curve behaves in a similar fashion to the cereals-only model of ref. [21] but also accounts for the increase in poverty observed around $t = 1966$ as seen in the alternative poverty index measures: the Head Count Index, Poverty Gap, and Squared Poverty Gap. These alternative measures have been scaled against the maximum of the index values. The model was constructed using data from 1959–1991 and verified against data in 2001 and 2005, each with more than 90% accuracy.

required implementation of appropriate multivariate tools for such assessment. In terms of integrating the concept of poverty in existing ABMs, there still exists a big gap. A self-consistent agreement between fixed-income-based poverty lines (impacting cognitive functions[36]), or income with subsistence production[37,38], or CD-based poverty lines[23] based on nutritional indicators is still missing. They also suffer from a disconnect with the structural aspects of asset and commodity markets whose outcomes generate poverty. It is this disconnect that precludes the use of those measures for anticipating future trends in poverty and arresting any expected increase in poverty through policy intervention. As the levels of development increase the focus of poverty has shifted from basic need food to other dimensions. This resulted in mushrooming literature on MPIs. Those indices also suffer from the use of arbitrary poverty lines, one for each commodity, and combine multiple dimensions into one by an index formula that is also often arbitrary.

In this article, we offer an alternative formulation of inequality based on the consumption-based poverty measure that is multi-dimensional in nature. We avoid using any arbitrary and exogenous information such as a poverty line or an exogenously specified mathematical formula for the poverty index. Instead, we use economic theory suggested CD, combined with objectively defined machine learning tools reducing the multidimensional poverty as a single dimensional measure of poverty index. We link the multidimensional poverty measure to the outcomes of asset market and commodity markets. We also model agent-based market exchange mechanism. We use extensive data over three decades from India to demonstrate how well our model works and how well it reproduces the trends in poverty in India. A traditional mechanism of ensuring veracity of a data-based model is to split the sample into two subsamples, one to estimate the model and another to predict and compare against the data points that were not considered in defining the initial continuum measure. That is a technique normally employed when the sample relies on cross-sectional data. In time series data analysis, it is not necessary, in fact may even be technically incorrect as time series data analysis tends to ascribe greater weight to data from the recent past (Markovian nature). In such context, splitting the sample might actually loose the information needed for better prediction. To confirm the robustness of our analysis, we replicate these methods with another data set (US data—Supplementary Information) to demonstrate that our observations are not just a statistical coincidence but are replicable in other country contexts.

Our model highlights two key aspects of inequality perception. First, multivariate construction is a key component of economic data analysis, implying all modes of income and expenditure need to be considered to arrive at a proper weighted estimate of poverty. Second, in developmental economics, machine learning as a tool functions better when integrated with statistical mechanics-based models for the purpose of economic prediction. This will explain why a reasonable body of literature dealing exclusively with artificial intelligence modes of data modeling have underperformed when it came to modeling poverty. More detailed statistical error measures, like Type I and II errors, p-values, and others could have been added; but this will not add anything to our conclusions and hence outside the scope of the present paper. It is possible to build a bootstrap element into model estimation to generate multiple replications of the sample to generate the standard errors and confidence intervals which is something that we intend to more extensive future analysis with statistically large enough samples where that will add further dimensions to error modeling.

Multidimensional poverty measures in India are few and far between[12,35]. By removing arbitrary functional forms and poverty thresholds and linking poverty measurement to structural agent-based economic models, our approach to poverty measurement and prediction goes a long way in advancing economic science associated with measurement of poverty. Further emphasizing the need for multivariate modeling[12,13], our analysis highlights how estimations based on individual PDFs could be drastically different from their combined conditional distribution (Fig. 2). The veracity of the conclusions drawn using Indian data are confirmed against US data (also from the[26] website, detailed in the Supplementary Information) with the dynamical PDFs (Fig. S1), their (Pareto distribution) rescaled counterparts (Fig. S2), consistent scaling (Fig. S3), and a time evolving dynamics of the Pareto exponent (Fig. S4) all conforming to the key signatures noted above. Both data modeling conclusions (India and the US) relate to the fact that the total poverty index calculated from the unified (multivariate) distribution shows a comparatively lower value than that for a cereal based poverty-index. These results, especially the latter, are not simple artifacts, rather they depict the comparative push-and-pull type economic dynamics between different modes of wealth flow that manage to stabilize (or destabilize) an economy, paving the way for social engineering.

## Methods

Engel law or Engel curve play a central role in our model. Engel curve is the observed relation between the consumption expenditure on any specific commodity and total income[15]. Economic agents, given their income, would like to choose a level of consumption that is the community norm as reflected by the point on the Engel curve corresponding to their income. We take the observed Engel curve derived from long-term time series data as a proxy for the perception people have of the relation between income and expenditure by people of a community. In a dynamical model, the economic agent strives to move up the income ladder and improve the level of living. The Engel curves of consumption expenses are concave toward their income axis, as in Fig. 1. In a hierarchical setup, they are concave with respect to the horizontal axis represented by the residual income left after meeting the more important needs as described in Sitaramam et al. Details related to sample sizes and data collection methods are all provided in ref. [26]. The model exclusively uses Indian data but is generic in its construct.

For a three-dimensional multivariate expenditure distribution, such steady-state Engel representation as in Fig. 1 of the statistics show the following data; here $\alpha$ is the non-dimensional scaling exponent representing the Malthusian saturation point beyond which more income will not translate in to more food consumption.

The analytical representation of the scaling exponent $\alpha$ comes from Eq. (13) that shows a Pareto decay for relatively larger values of expenditure. In our multivariate formulation of income–expenditure statistics, $\alpha$ is the consummate multivariate representation of the poverty index that we provide in more details hereafter. The comparison between model outputs and real data have all been benchmarked within $\pm\sigma$, where $\sigma$ is the standard deviation measured from real data (also provided explicitly in ref. [26]).

**Machine learning: Dimension Reduction**. Dimension reduction is the process of mapping consumption expenditure data, $y_i \in \mathbb{R}^m$, to a (typically) lower dimensional space, $z_i \in \mathbb{R}^n$, with $n < m$ such that it can be analyzed. Many algorithms exist for the projection of data to a lower dimensional space[32]. Perhaps the most well-known algorithm is PCA effecting a linear transformation upon data $y_i$. PCA identifies the optimal projection only where the manifold upon which the true data, $y_i$, sits is linear and Euclidean and as such is incapable of reliably analyzing real-world processes. The notion of a global Euclidean chart was improved upon in refs. [30,33] resulting in the LLE algorithm. The patchwork of local Euclidean neighborhoods attempts to linearly map data preserving local neighborhoods; however, the degree of nonlinearity in the real world often leads to the failure of LLE in preserving the topological ordering of data.

Tenenbaum et al.[31] use a global approach where a local Euclidean chart is assumed and non-neighborhood distances are calculated through use of a geodesic known as IsoMap. It should be noted that the local chart can be non-Euclidean where an alternative dissimilarity measure is specified. Once the local distances are calculated, the global distances between non-neighboring points approximated using Djikstra's algorithm. The resulting dissimilarity matrix is then linearly embedded into an $n$- dimensional space in a similar fashion to PCA. Despite the more accurate approximation of global distances in IsoMap compared to PCA, the linear mapping can still result in a low-dimensional embedding, but with poor neighborhood preservation. The advantage of the linear embedding used in the IsoMap algorithm is that a global minimum of the linear embedding is found, in contrast to other nonlinear dimension reduction algorithms requiring gradient-based optimization of non-convex cost functions. However, in some applications,

the ease of embedding in IsoMap does not overcome the limitations of the linear process.

The approach given in ref. [27], called NeuroScale (NS), presents a nonlinear mapping from $\mathbb{R}^m \to \mathbb{R}^n$ based on the Sammon mapping[34]. The latent points, $z_i$, are identified through minimization of a mapping cost function known as STRESS:

$$E = \sum_{i,j} \frac{\left( D_y(i,j) - D_z(i,j) \right)^2}{D_y(i,j)}, \tag{4}$$

where $D_y(i,j)$ represents the dissimilarities between datapoints $y_i$ and $y_j$ and $D_z(i,j)$ represents the dissimilarities between the dimension reduced points $z_i$ and $z_j$. Typically the Euclidean distance is used in the dimension reduced space such that $D_z(i,j) = \|z_i - z_j\|$, resulting in a Euclidean embedding of the data $y_i$. The observation dissimilarities $D_y(i,j)$ are application specific and should be selected based upon the nature of the observations, for instance where $y_i$ consists of binary elements, a binary distance measure should be used instead of the Euclidean distance which is often implemented. The latent points, $z_i$, are identified through minimization of Eq. (4) using a form of gradient descent known as Shadow Targets[29]. In addition to the nonlinear mapping, a further benefit of NS over the linear mappings discussed above is that it effects a parameterized mapping using a Radial Basis Function Neural Network[39], such that new observations $y^*$ can be projected to the dimension-reduced manifold. CCA[28] is similar to NS with a slight modification of the STRESS term from that of Eq. (4). This modification includes a nonlinear weighting, $F(D_y(i,j), \lambda)$, of non-neighboring points, approximating their distances in a non-Euclidean fashion as Isomap does:

$$E = \sum_{i,j} \left( D_y(i,j) - D_z(i,j) \right)^2 F(D_y(i,j), \lambda). \tag{5}$$

Perhaps the most popular weighting scheme is to neglect distances associated with far points offering a nonlinear local projection known as step-CCA. Global mappings of CCA require the selection of a parameterized nonlinear weighting function which can have a large impact on the resulting embedding and as such we choose to implement step-CCA with a neighborhood width allowing for the creation of a fully connected neighborhood graph.

**Agent-based model.** We now present an agent-based barter model of trade to capture the complex behavior of interacting economic agents in an environment of market institutions and social interactions.

We assume that the economic agents interact with each other in an institutional environment with markets for commodities and assets, and with the non-market institutions such as the State and other non-profit institutions. In the labor markets, agents exchange their human skills (work) for wages and salaries. In the commodity markets, agents exchange money for commodities. In asset markets, agents exchange their physical and financial capital for financial returns. There are those who only have human capital to exchange but no financial assets. Such people get only wages and salaries, and spend most of it and save little for contingencies. Others, who have other assets other than human capital, save after meeting consumption needs. Agents with only low wage income exchange their attributes of low-income destitution or deprivation for entitlements for grants or financial assistance. In line with Ernst Engels, our focus will be on these two sets of people who depend only on their human skills and save nothing, and those whose destitution is such that they exchange the destitution attributes for grants and other financial assistance.

ABMs are of two kinds: (i) based on an optimization model depicting the self-interest goal of agents or (ii) based on a behavioral model using certain rules of thumb. It was observed that behavioral models tend to do well, apart from in exceptional or contrived conditions. Aumann[40] takes a synthetic view and shows that the exceptions occur as the behavioral models do not explain how the rules evolve. Thus, by explaining how the behavior rules evolve, one can get a synthetic model that is better than either. Whether it is an equation-based model or a behavioral model, one can specify it at any degree of aggregation. The more disaggregated the model is, the larger would be the size of the number of parameters and larger would be the computational inaccuracy. It is for this reason that ABMs are used at nano level by physicists and computer scientists to build econophysics models involving computer simulations[41,42].

The demand for commodity by an individual $i$ at time $t$ is given by typical arguments of Walrasian multiple market temporary equilibrium, as developed in ref. [43]. According to Grandmont[43], the next period consumer equilibrium will be a time adjustment from the previous period equilibrium with an increase or decrease in the quantity exchanged, taking into account the common information about the markets in the next period ($I_t$) as well as specific information that agent $i$ has for the next period ($I_{it}$)

$$x_{it} = x_{it-1} + \triangle x_{it-1}(I_t) + \triangle x_{it-1}(I_{it}), \tag{6}$$

where $I_t$ refers to the information commonly available to all agents in period time $t$, while $I_{it}$ is the information in period $t$ specific to agent $i$; $x_{it}$ is a vector representing all consumer goods purchased by individual $i$ at time $t$, and it depends on the common market prices faced by all agents and income of the agent $i$. The incremental changes depend on the expectations and their realizations through a groping process of adjustment of actuals with expected. As expectations are

adaptive and depend on the actual and realized values of the previous period, we can rewrite parts of the above equation as

$$\triangle x_{it-1}(I_t) = g_1(x_{it-1}) + x_{it-1}\omega_t, \tag{7}$$

where $\omega_t$ is a general market disturbance that affects equally all economic agents, while the variance of the market disturbance is proportional to the square of the level of economic activity in the previous period ($x_{t-1}$). Likewise, we write:

$$\triangle x_{it-1}(I_{it}) = g_2(x_{it-1}) + x_{it-1}\epsilon_{it}, \tag{8}$$

where $\epsilon_{it}$ depicts how the $i$th economic agent deviates from the others in the same market. Plugging these expressions into the previous equation, we get:

$$x_{it} = x_{it-1} + g(x_{it}{}^e) + x_{it-1}\,\eta_{it}, \tag{9}$$

where $\eta_{it} = \omega_t + \epsilon_{it}$, $g$ is appropriately defined and depends on $g_1$ and $g_2$, $x_{it}{}^e$ is the expected or anticipated quantity at period $t$ for agent $i$, and $\eta_{it}$ is a random variable derived from a Gaussian distribution (under the assumption that the privileged private information of the agents' is distributed likewise). This distribution can be modified if necessary. Similar arguments apply for the agents working on the supply side. Our model can be easily extended to build a macroeconomic model by assuming that the individual equilibrium demands aggregate to market demand and individual equilibrium supplies are aggregated to market supply. In a temporary stochastic equilibrium model such as this, we define the market equilibrium as a stochastic Markov equilibrium as defined in ref. [44]. We then assume that demand and supply are co-integrated over time. They adjust with each other through a dis-equilibrium dynamics mechanism, giving rise to an error-correction.

Applying the above temporary equilibrium concepts to the labor and asset markets, we get the following income generation model:

$$y_{it} = y_{it-1} + g(y_{it}{}^e) + y_{it-1}\,\eta_{it}, \tag{10}$$

where $y$ is rescaled $x$. Assuming that these markets work on daily basis or weekly basis, one can take the daily version of it and that daily change is infinitesimal compared to yearly data points we have in our data base. This then leads to the Langevin and complementary Fokker–Planck formulation in line with ref. [23].

**Langevin Fokker–Planck model.** The CD function $CD(y_i, t)$ measures the amount of income lacking in an income class $i$ that is needed to reach the saturation level of consumption of the essential commodity. This was well explained and estimated in some previous works[15,16,21,23]. But there is a fundamental knowledge gap in all such previous measures of the CD function in that they relied only on the key expenditure variable, cereals, as the unique descriptor of CD. The tacit assumption was that the other expenditure categories, even though contributing to consumption and welfare, will all be subsumed within the inequality outliers defined by the dominant cereal-expenditure. As Figures 3a and 3b clearly demonstrate, this was a weak foundation to base any model on, since intrinsically each of the three expenditure modes, cereals, other foods, and non-foods, abide completely different distribution functions. In other words, any self-consistent measure of poverty needs to analyze the multivariation of all contributing expenditure modes. Using tools from the machine learning literature[33], we have combined all three contributing expense modes into a one-dimensional reduced variable that combines information about their individual and correlated dynamics. As shown in Figure 3a, 3b and 4b, the choice of methods (Neuroscale, LLE, IsoMap, CCA, PCA) barely affects the qualitative distribution.

Following on from the ABM, each agent spends an income $y_i(t)$ from wages and salaries and asset markets and expect for the next period an income of $y_i^e(t) > y_i(t)$, only a fraction $\beta$ from that is immediately accrued. The agent $i$ spends a fraction $\alpha$ of savings or borrowing, depending on the sign of the expression in the parenthesis, as either interest expense or as cost of holding assets (for simplicity we are assuming the cost of lending and borrowing are the same and equals). The resultant stochastic (Langevin) model that we then get from this construction is as follows:

$$\frac{dy_i}{dt} = \beta\,y_i^e - \alpha\,(y_i - CD_i) + y_i\,\eta_i(t), \tag{11a}$$

$$<\eta_i(t)\eta_j(t')> = D_0\delta(t - t')\delta_{ij}, \tag{11b}$$

where $\beta\,y_i^e$ is the maximum attainable level of income (proportional to the mean growth rate $y_i^e$ of income) at time $t$ for the $i$th class of individuals. The CD function cumulatively represents all three nodal expenditures, cereal, other-food, and non-food, whose non-dimensional multivariate form defines our new variable $C(y)$. In other words, $C(y)$ is a suitably weighted function of $C_1(y)$, $C_2(y)$, and $C_3(y)$. For reasons of brevity, we will deliberately obfuscate the income class index $i$ from all future descriptions as it is only a generic dummy index.

The Langevin model defined in Eq. (11a) leads to the following Fokker–Planck equation[45] that shows the time variation of the income distribution function:

$$\frac{\partial \hat{f}}{\partial t}(y, t) = \frac{\partial}{\partial y}\left\{ [(\alpha + 2)y - C(t) - CD(y, t)]\,\hat{f} + y^2\,\frac{\partial \hat{f}}{\partial y} \right\}. \tag{12}$$

The coupled dynamics described above, involving the probability density function of income $f(y, t)$ and CD function $CD(y, t)$, represents the fact that

**Table 1 Decay values, $\alpha$, for each of the original data dimensions and the dimension reduced models governing the decay of consumption deprivation, CD($y$), in Fig. 2.**

| Data/Model | CD($y$) Decay rate, $\alpha + 2$ |
|---|---|
| Cereals | 3.5663 |
| Other food | 1.4853 |
| Non-food | 0.81851 |
| Multivariate-NeuroScale | 1.0739 |
| Multivariate-LLE | 0.79709 |
| Multivariate-IsoMap | 1.0425 |
| Multivariate-CCA | 0.95466 |
| Multivariate-PCA | 0.95739 |

The global mappings of IsoMap and NeuroScale show an expansion in the decay in contrast to the multivariate exponents calculated using LLE, CCA, and PCA. Our results show that the multivariate exponent value ($\alpha + 2$), averaged over all five dimensional reduction measures, is ~–1.0, as opposed to –3.5 for a cereal-consumption-based exponent[21].

effective trade in assets ensues only when $y > \text{CD}$, as discussed before (the solution to the Burgers equation is shown in Eq. (14)).

In the steady state, $\overline{C}(t) = C_0$ and $\text{CD}(y) = \frac{V_0 K_0}{K_0 + y}$ ($t \to \infty$ limit), which gives us the steady-state income distribution:

$$f(\hat{y})_{t \to \infty} \propto \frac{e^{-\frac{(C_0 + V_0)}{y}}}{y^{\alpha + 2}} \left(1 + \frac{K_0}{y}\right)^{V_0/K_0}, \qquad (13)$$

where $V_0$ and $K_0$ represent values of $V(t)$ and $K(t)$ on specific years (represented by $t$). Figure 7 uses a linear approximation in extrapolating the data points for $V$ and $K$ as functions of time $t$. Note such an extrapolation does not restrict the remit of this model as with a simple Markovian approximation, the value of these parameters at time $t$ can be used to predict the value at the following time point $t + 1$. We have checked this and can confirm that the results do not differ significantly. We observe from the scatter diagram that there is a piecewise linear relation, a linear function until year 12 and another after year 12. As we are using only the data for the latter period, the fit is linear with perfectly respectable fit. The proportionality constant can be evaluated from the condition $\int_{y_0}^{\infty} \hat{f}(y, t \to \infty)\, dy = 1$.

The parameter $\alpha$ is highly country specific but is reasonably stable over the range of years of data analyzed. As shown in Table 1 above, the mean multivariate-$\alpha$ value is ~–1.0. As previously shown in refs. [21,23], Eq. (12) admits of a closed form harmonic solution with a hypergeometric function as its generating function.

**CD dynamics.** To study an agent-based version of the CD model, we follow the prescription as in ref. [23]. We consider an agent $i$ who has (multivariate) expenditure $y$ line at time $t$. Trade is entertained only when $y > \text{CD}(y, t)$ at all times. Our interest is in a trade situation where an amount $\Delta y$ has been transacted across agents belonging to two income classes.

In order to neutralize the economic disturbance, this extra source needs to be isotropically diffused between economic neighbors of the two agents responsible for this exchange; this necessitates a diffusion term in the dynamics: $\nu(t) \frac{\partial^2}{\partial y^2} \text{CD}(y, t)$ ($\nu(t)$: time-dependent diffusion constant). This economic palliation will be challenged by further wealth accumulation that will happen as a function of the rate of change of CD in between these income classes with respect to the combined (i.e. multivariate) expenditure $y$, calibrated against CD itself: $\text{CD}(y, t) \frac{\partial}{\partial y} \text{CD}(y, t)$. Together, the income distribution time variation model is then seen as follows:

$$\frac{\partial}{\partial t} \text{CD}(y, t) + \text{CD}(y, t) \frac{\partial}{\partial y} \text{CD}(y, t) = \nu(t) \frac{\partial^2}{\partial y^2} \text{CD}(y, t),$$
$$\frac{\partial}{\partial t} \text{CD}(y, t) = V(t) K(t) \frac{2\nu(t) + V(t) K(t)}{(K(t) + y)^3}. \qquad (14)$$

The trendline fits shown in Fig. 6 are for years beyond 1971. Evaluation before that can be done but is redundant as the new base market policy against which our evaluations are all calibrated started only in 1971. These are linear regression fits only as a first approximation. Better nonlinear fits can be attained but that does not have any measurable impact on the outcome.

In Fig. 6 above, the $x$-axis uses round numbers depicting timelines, rather than year numbers; this is to allay the aperiodic nature of data collection over the years. A linear regression fit to these time-dependent variables shows the following: $V(t) = 77.61 - 0.83t$ and $K(t) = 54.71 - 1.15t$. Equation (14) represents a form of the celebrated Burgers equation in fluid mechanics[46] that characteristically admits of shock waves, an obvious economic after turn. For a real-time varying model, Eq. (14) can only be solved numerically utilizing the linear equations for $V(t)$ and $K(t)$.

## Data availability

All relevant data have been accessed from the World Bank repository (cited in ref. [26]) and the codes used for simulation, leading to the plots produced in this work, can be accessed from the Aston University repository (https://researchdata.aston.ac.uk) through the following doi-link: https://doi.org/10.17036/researchdata.aston.ac.uk.00000469.

## Code availability

Codes solving the stochastic model and subsequent Machine Learning analysis were all written in Matlab, which could be accessed from the Aston University Library website (https://researchdata.aston.ac.uk) through the following doi-link: https://doi.org/10.17036/researchdata.aston.ac.uk.00000469.

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

## Acknowledgements
A.K.C. acknowledges R. Badawy and E. Grela for their careful perusal of the manuscript.

## Author contributions
A.K.C. led the project by conceiving the core scientific idea and model; T.K.K. provided the economic ramification of the mathematical model and key validation of the model using alternative data sources; I.R. conducted the initial machine learning-based data modeling. All authors contributed equally to the writing of the manuscript.

## Competing interests
The authors declare no competing interests.
