## [Peer Review File · Nature Communications]

Reviewers' comments:

Reviewer #1 (Remarks to the Author):

This paper offers results from an interesting modelling exercise on poverty. The multi-method approach is particularly interesting. However, the connection between the stats-based development of differential equation and the agent-based approach makes little sense to me. Typically, agent-based models represent systems in a highly disaggregated manner and focus on the interaction between human entities (and often human-environment interactions). Introducing differential equations as behavioural response functions contradicts the very essence of such complex systems thinking (see Gilbert 2008 and many other publications for this point). I understand that from an econo-physics perspective much potential lies in the combination of differential equations and the rules-based approach of agent-based models. And I'm not opposing all solutions that have emerged over the years. But I find it very hard to justify for this particular context. Much of the existing literature on multidimensional definitions of poverty look at individuals or households. There is no evidence that individuals or households follow such continuous response functions. Instead, the structure of such response functions has very different shapes. I suggest to read some of the literature on how to parameterize empirical agent-based models for this matter. The logic presented here is more relevant for large cohorts of households. But then the argument of multidimensionality of poverty breaks down because the poverty of larger population segments depends on many aspects, including their subsistence production, which is not finding any consideration in this paper. I find the first steps of this paper useful and would find it interesting how it compares with all the existing multidimensional poverty indices that already exist (there are many!). If agent-based methodology is critical I suggest building on (existing) disaggregated designs instead of the aggregated definition and, thereby, moving from a continuous differential equations as response functions to discrete logic based response functions to avoid the inherent methodological contradiction to represent human behavior.

Reviewer #2 (Remarks to the Author):

Synopsis

I find this paper very interesting as it attempts to incorporate quintessentially economic concept of poverty into an emerging body of literature of machine learning. The paper claims to offer a new model that combines three groups of food expenditure (basic food, other food and non-food) to measure and forecast poverty/ inequality using an Indian dataset. The main contribution of the study is -providing a multivariate construct for poverty line using machine learning tool. I am sure that this novel attempt will be of interest to the economists in general and scholars and agencies dealing poverty analysis in particular.

I have a few comments regarding the paper.

Comment 1

I do not see that the authors have taken a sufficient effort to provide appropriate literature that deals with the poverty line estimates and forecasts. While the authors try to provide a rationalization of the study and its method what is missing here is a systematic survey of extant literature that could lead the reader to identify the importance of the study. I suggest to include a section or significantly expand and strengthen the rationale presented in the introduction section with related literature.

Comment 2

While I am not in a position to speak/ comment explicitly about technical details of machine learning, I think employing the method offers novelty to the study. However, my concern is – why emphasis placed on describing the methodology overweighs the contextual relevance /appropriateness of the

study? Isn't the contextual relevance that is of interest to a wider group of readers?

Comment 3

Regarding the important results of the study, (the multivariate poverty index) I like to see more discussion of results- In particular fig 5. For eg-The Indian poverty levels at national level) have significantly decreased after 1970 – due to what major events/factors?

Comment 4

I would like to see elucidation concluding remarks section that provide a contextual picture to the reader. The main question that I was looking for answers are simple- How good/ better the new poverty line compared to existing ones? If that outperforms existing ones what could be the policy relevant implications?

To address these comments I suggest the authors to undertake a comprehensive rewrite- specially the introduction and concluding sections.

Editorial

Choi and Moneta (2010) is not in the text but in the reference list –The correct reference is Chai and Moneta (2010)

Reviewer #3 (Remarks to the Author):

Overview:

All previous approaches have considered only one mode of expenditure which is basic food (cereals) as a unique descriptor of economic inequality. Since each of three expenditure modes basic food (cereals), other food and non-food abide completely different distribution function, so for any self-consistent measure of poverty need to analyze based on the contribution of all three modes of expenditure. The proposed work includes all three modes cereal, other food and non-food for measuring and analyzing the poverty.

Positive:

1- This an important problem to define the poverty index, it is not just for a social purpose but also for the policymakers. The paper raises an important issue that poverty cannot be explained by just a single factor there should be other dimensions too. This will gives the more robust predictions. The author proposed a method that considers the poverty index prediction as a multivariate problem. They shows the linear/nonlinear dimension reduction method can model this problem more robustly. Later we can include more dimension in the model if we find it is also an essential factor for the poverty index prediction.

Negative and Comments:

There is not much novelty in terms of approach. It can be impactful for society or researcher. But from the Machine Learning perspective novelty is limited.

1: A linear model (linear regression) has been fitted on the data for two variables $V(t)$ and $K(t)$. Which work as the threshold line for the measuring the poverty. Whereas in the real scenario the data distribution is highly nonlinear, so a nonlinear model (higher degree polynomial) is expected to fit on the data for both the variables $V(t)$ and $K(t)$.

2: Since there is three mode of expenditure cereal, other food and non-food. The comparison has

been shown only cereals vs. multivariate poverty index. What about other food vs. multivariate poverty index and non-food vs. multivariate poverty index, expected these comparisons also.

3: How to choose the number of reduced dimension in PCA etc.?

Performance should be shown for the different number of reduced dimension for proving the fact and conclusion better.

4: Need a better explanation (ablation study) to conclude that how three modes of expenditure cereals, other food and non-food effect the prediction of poverty.

5- In the NeuroScale in (Eq-1) it is just a linear mapping it will be nonlinear only if you are using some nonlinear function mapping. I think something is missing. Here you need some kernel map for the nonlinear mapping.

6- What will be the behavior of the proposed approach if you are doing the extrapolation of the poverty index, i.e., beyond the given data period (1959-1991). For this, you can hide some year's data from the dataset use the rest of data for the training and use the hide data for testing so that we can see the result of the proposed method in case of extrapolation.

7- In the mentioned dimension reduction technique, we can not give the confidence score about our predicted value. But in the sensitive scenario with the predicted score if we can also tell how much confidence we have about our prediction it will be a better scenario. Also, It will be more helpful in the case of extrapolation. Here I would like to see the result of such method that can give the confidence score also, e.g. Bayesian regression(linear/nonlinear). Please refer to "Christopher M. Bishop: Pattern Recognition and Machine Learning, Chapter 7.2.1."

8- I think in all the three scenarios (cereals, other food and non-food), the cereals are the most critical factor, while in the Appendix for fig-6 it is given that PCA has the weight allocation to each scenario 0.1694, 0.4336 and 0.8851, for cereals, other food and non-food respectively. This clearly shows that the non-food has the highest weight to decide the poverty index. While intuitively cereals should have the highest importance. Please justify the above weight obtained by the model.

9- Also the mentioned approach is not robust for the outliers, and a few noisy samples can affect the whole model. Can you show some result using the robust regression technique? Please refer to "<http://www.eng.tau.ac.il/~bengal/outlier.pdf>."

10- Is there any other dataset that can be used also to show the result?. Most of the time the ML algorithm that works fine on the one dataset fails to another one.

Note: Please consider this comment as major revision.

Thanks

Referee 1

Comment 1: The connection between the stats-based development of differential equation and the agent-based approach makes little sense to me. Introducing differential equations as behavioral response functions contradicts the very essence of such complex systems thinking (see Gilbert 2008 and many other publications for this point). Much of the existing literature on multidimensional definitions of poverty look at individuals or households. There is no evidence that individuals or households follow such continuous response functions.

Our response:

First of all, we appreciate the referees' perceptive observations and constructive critical comments and his apparent appreciation for the new trend in applying scientific computer aided analytic tools to address various complex social phenomena. One of them is poverty and that is what we address in this paper. However, most of research on poverty and multi-dimensional deprivation have used arbitrary and subjectively chosen poverty line. Our approach calls for its revision and its replacement by a data driven market determined threshold where economic agents do not save and spend all their income.

We start with an agent-based model and then use individual economic agents to explain our model (theory building needs a scientific construct that is both deductive and inductive, employing Nigel Gilbert's terminology). We assume that the economic agents interact with each other in an institutional environment with markets for commodities and assets on the one hand, and the non-market institutions such as the State and other non-profit institutions on the other. In asset markets agents exchange their human capital and physical capital for financial returns. There are those who only have human capital to exchange but no financial assets. Such people get only wages and salaries and many of them spend most of it and save little. Others, who have other assets other than human capital, save after meeting consumption needs. In the latter two types of institutions (State and non-profit) people endowed with only human capital exchange their attributes on low-income destitution or deprivation for entitlements for grants or financial assistance. Ernst Engels focused on, and so do we, on these two sets of people those who depend only on their human skills and save nothing, and those whose destitution is such that they exchange the destitution attributes for grants and other financial assistance.

There are two types of equilibrium concepts in economics. One is market equilibrium that is an aggregate concept. The other is a consumer equilibrium and producer equilibrium. These relate to the optimal solutions to optimizing behavior of the economic agents, characterizing a neoclassical assumption of a rational economic agent. Our basic behavioral assumption is that individuals observe the market and adapt their behavior from period to period. The demand for commodity by individual i at time t is given by typical arguments of "temporary equilibrium" (see J.M. Grandmont, 1977) the next period equilibrium will be a time adjustment from the previous period equilibrium with an increase or decrease in quantity exchanged, taking into account the common information about the markets as well as specific information that agent i has for the next period:

$$x_{it} = x_{it-1} + \Delta x_{it-1}(I_t) + \Delta x_{it-1}(I_{it})$$

Where I_t refers to the information commonly available to all agents in period t , while I_{it} is the information in period t specific to agent i . x_{it} is a vector and it depends on the common market prices faced by all agents and income of the agent i . The incremental changes depend on the expectations and their realizations through a groping process of adjustment of actuals with expected. Without loss of generality we can rewrite the above equation as:

$$x_{it} = x_{it-1} + g(x_{it}^e) + f(x_{it-1}) + \eta_{it}$$

where x_{it}^e is expected or anticipated quantity at period t for agent I , and η_{it} is a random variable having a Gaussian distribution (under the assumption that the privileged private information agents is distributed likewise. This distribution can be modified if necessary. Similar arguments apply for the agents working on the supply side, and under equilibrium demand is equal to supply, or they adjust to each other through a nonequilibrium dynamics. Applying this to the asset market we get:

$$y_{it} = y_{it-1} + g(y_{it}^e) + f(y_{it-1}) + \eta_{it}$$

Assuming that these asset markets work on daily basis or weekly basis or at most monthly basis, one can take the daily version of it and assume that daily change is infinitesimal compared to an yearly data points we have in our data base. Hence, we approximate the finite difference by its continuous equivalent to arrive at our Langevin equation:

$$\frac{dy_i}{dt} = \beta Y_i(t) - \alpha(y_i - CD_i) + \eta_i(t),$$

$$\langle \eta_i(t) \eta_j(t') \rangle = D_0 \delta(t - t') \delta_{ij}.$$

Do note that, a continuum approach towards modeling and continuum response function are very different theoretical structures. Ours belong to the former category, not the later.

Comment 2: The logic presented here is more relevant for large cohorts of households. But then the argument of multidimensionality of poverty breaks down because the poverty of larger population segments depends on many aspects, including their subsistence production, which is not finding any consideration in this paper.

Our response:

As explained above, we do start with an agent-based model describing the individual economic behavior by treating such agents as micro units in a large ensemble of interacting agents, and hence the Physics based analogy. As described above the model captures asset market solution leading to destitution where poor unskilled labors are engaged in subsistence production and get in exchange only a very low income not enough to meet its consumption needs. The referee may note that this aspect is now more clearly explained in the revised draft.

Comment 3: I find the first steps of this paper useful and would find it interesting how it compares with all the existing multidimensional poverty indices that already exist (there are many!). If agent-based methodology is critical I suggest building on (existing) disaggregated designs instead of the aggregated definition and, thereby, moving from continuous differential

equations as response functions to discrete logic-based response functions to avoid the inherent methodological contradiction to represent human behavior.

Our response:

Our theoretical structure incorporates exactly those vital threads that the referee is asking for. Our agent-based exchange model is a disaggregated model at individual household level and the data we used is also at individual household level from household sample surveys conducted in India over several decades. We start a discrete version of market temporary equilibrium and move to its approximation by a continuous differential equation version for analytic rigor and simplicity.

There are two types of multi-dimensional deprivation indices. Both arise from a famous study by Sudhir Anand and Amartya Sen for UNDP (1994). These are multi-dimensional poverty indices of Alkiere and Foster (2007) or Human Development Indices arising from UNDP's efforts. The former uses several dimensions of poverty and for each one of them it uses its own poverty line. We object to such use of arbitrary poverty lines. Thus, we prefer somewhat the Human Development Index of UNDP. However, we suggest replacing the all-inclusive Human Development Index by a new multi-dimensional poverty index that confines to basic essential needs of destitute population defined as those who spend all their income on consumption and have no savings left. The novelty of our approach is that we generate an aggregate index from the constituent basic needs employing machine learning methods to extract the nonlinear principal component. We do not use any arbitrary mathematical formula to define the index.

References:

J.M. Grandmont (1977), "Temporary General Equilibrium Theory", *Econometrica*, Vol. 45, No. 3, April, pp. 535-572.

Sudhir Anand and Amartya Sen (1994), Sustainable **Human Development: Concepts and Priorities**, *UNDP Human Development Report Office 1994 Occasional Papers*.

Alkiere, Sabina, and James Foster, (2015), *Multi-Dimensional Poverty Measurement and Analysis*, Oxford University Press.

Referee 2

Comment 1: I do not see that the authors have taken a sufficient effort to provide appropriate literature that deals with the poverty line estimates and forecasts. While the authors try to provide a rationalization of the study and its method what is missing here is a systematic survey of extant literature that could lead the reader to identify the importance of the study. I suggest to include a section or significantly expand and strengthen the rationale presented in the introduction section with related literature

Our response:

We believe that our new modeling opens up an alternative formulation of inequality modeling which, given its entrenchment in data-based outliers, is closer to econometrics than the available literature on poverty modeling. An abject lack of such independent modeling approach in the existing poverty literature precludes essential literature that we could cite, which hopefully explains the reason why we did not deal with extensive literature on poverty studies with poverty

lines. Our approach departs from the existing poverty studies significantly by rejecting the need and use of a poverty line. Along with the rejection of the poverty line goes the need to cover that literature, as the limited journal space can be better utilized more effectively to describe our approach.

It is however correctly pointed out that our introduction should more explicitly explain (and identify) the importance of our approach. This is being done in our revised version of the paper.

Comment 2: While I am not in a position to speak or comment explicitly about technical details of machine learning, I think employing the method offers novelty to the study. However, my concern is – why emphasis placed on describing the methodology overweighs the contextual relevance /appropriateness of the study? Isn't the contextual relevance that is of interest to a wider group of readers?

Response:

We appreciate these concerns and hence are thoroughly revising the style of presentation. We are now explaining in the revised version the importance of using the machine learning approach to derive the multi-dimensional index by citing the literature on multi-dimensional poverty and Human Development Index and their limitations to identify and emphasize the need to use the machine learning methods. *We are shifting the technical details to a longer methodology section, as suggested by the editor.*

Comment 3: Regarding the important results of the study, (the multivariate poverty index) I like to see more discussion of results- In particular fig 5. For eg-The Indian poverty levels at national level) have significantly decreased after 1970 – due to what major events/factors?

Response:

We appreciate this comment. In fact, in our effort to limit the size of the paper, we had to truncate discussions on some important aspects, this being one of them. We did deal with this economic explanation the referee is asking in our earlier paper in the Journal of Development Studies. In fact, the Indian Development Planning experience emphasized growth until the Fifth Five Year Plan that started in 1971. From the Fifth Five Year Plan, the emphasis had shifted to poverty eradication and growth with equity. Our poverty estimates vindicate the shift in policy focus. This is now explained in the text.

Comment 4: I would like to see elucidation concluding remarks section that provide a contextual picture to the reader. The main question that I was looking for answers are simple- How good/ better the new poverty line compared to existing ones? If that outperforms existing ones what could be the policy relevant implications?

To address these comments I suggest the authors to undertake a comprehensive rewrite- specially the introduction and concluding sections.

Response:

This is a very justified comment and has now been addressed by a thorough rewrite of the Introduction and Conclusion, including parts of the inner text which have now been revamped with additional new materials (a new “Agent Based Models” section under “Methods and Methodology”) and plots. We have now extensively compared our poverty estimates with the existing estimates. We wish to emphasize that most of the existing poverty literature deals with alternate approaches differing either in the data used or using different poverty lines but using the same income-based poverty measure. We offer an entirely different way to define and measure poverty. Hence, strictly speaking they are not comparable as tracking the same object. We have also explained the contextual advantages of our approach. The fact that the referee is raising this issue raises a question in our mind “why our emphasis on the advantages of our approach is not being effective?” As the referee hinted in the concluding part of the comment, these main features should be broken into two parts. In part one, we must explain the reasons why we are using the approaches we chose. In the second part, we must explain if our approach did meet the needs for which we chose those approaches. The first part should go in the introduction while the second part should go to the concluding section. This is the approach that we have undertaken while revising our manuscript and we hope that the result justifies our effort.

We thank the referee for pointing out the omission of Choi and Moneta reference. We have added this in the revised version.

Referee 3

We sincerely appreciate the positive comments from this referee. We also wish to add that a key aspect of our work is the self-sufficiency of the poverty measure without any unwanted reliance on any arbitrarily chosen poverty threshold (poverty line) or a specific arbitrarily chosen functional form for combining the various dimensions into a single index. Our approach inherently and intrinsically relies on data that then guides the (unbiased) outcomes, as reflected by the agent-based market exchange model determine that point where the agents do not save, and the machine learning data reduction technique determine the way to combine the different dimensions into a single dimension. With the advanced multivariate analysis tools from machine learning, we could now decode the interdependence of the different expenditure modes and how such dependence contributes to the poverty index, *a first in the econometrics of inequality modeling*.

Comment 1: A linear model (linear regression) has been fitted on the data for two variables $V(t)$ and $K(t)$. Which work as the threshold line for the measuring the poverty. Whereas in the real scenario the data distribution is highly nonlinear, so a nonlinear model (higher degree polynomial) is expected to fit on the data for both the variables $V(t)$ and $K(t)$. Figure 7 uses a linear approximating in extrapolating the data points for V and K as functions of time t . Note such an extrapolation does not restrict the remit of this model as with a simple Markovian approximation, the value of these parameters at time t can be used to predict the value at the following time point $t + 1$. We have checked this and can confirm that the results do not differ significantly.

Response:

This is a correct observation and we have already addressed such a nonlinear functional extrapolation in our previous work (Ref 10). As shown there, the nonlinearity assumes very minor importance in the overall poverty measure. Hence, to stick to a minimalist description of linear regression of parameters, we have used the linear formulas. But, as one can see there is a regime shift that seems to coincide with the shift in development strategies followed in India. There was an increase in $V(t)$ and $K(t)$ until round 10 and then there is a secular decline in both $V(t)$ and $K(t)$. This corresponds to similar movements in the poverty estimates themselves, as in the new Figure 7. We are grateful to the referee for pointing this out and we have made the necessary change in our final draft.

Comment 2: Since there is three mode of expenditure cereal, other food and non-food. The comparison has been shown only for cereals vs. multivariate poverty index. What about other food vs. multivariate poverty index and non-food vs. multivariate poverty index, expected these comparisons also.

Response:

Our contribution is an extension of the model we presented earlier in Ref 10 that dealt with estimation of poverty in a single dimension using cereal deprivation. This extension is a dimensional reduction of a three-dimensional index. Hence, we compared our earlier index with current improved index. As cereals occupy a more essential need than other needs index based on that alone would be higher, as the moment people start consuming other commodities their poverty level decreases. However, to show the dependence on all three dimensions, and actually on their projections, we have now added 3 more plots in Figs 3a, 3b and 4. As explained in the revised text, the weighted average over all 3 expenditure modes show deviations from the individual probability distributions which have now been represented in Fig 3b.

Comment 3: How to choose the number of reduced dimension in PCA?.

Performance should be shown for the different number of reduced dimension for proving the fact and conclusion better.

Response:

The 'reduced dimension' in PCA, as also in other machine-learning protocols, is effectively a nondimensional weighted representation of all 3 dimensions combined with a certain 'dimensional loss' due to the linear projection on to the 3-dimensional manifold. Figure 4 compares the relative contributions using multidimensional visualization.

Ideally one should apply this procedure at more disaggregated level using a finely divided commodities or commodity groups, arranging them in hierarchies of need and then ask the question: What are the basic needs? Where do we stop in hierarchy of needs? Is there a fine division between the basic needs and other needs? Etc. These are outside the purview of this paper.

Comment 4: Need a better explanation (ablation study) to conclude that how three modes of expenditure cereals, other food and non-food effect the prediction of poverty.

Response:

As hopefully now explained through the figures, the multivariate distribution varies significantly from the individual distributions that assume other two variables unchanged. Poverty measures,

defined as consumption deprivations from saturation levels, that are derived from these distributions will also differ significantly between all three of them. When it comes to prediction, the prediction of the multidimensional poverty is the important one, which we have done. The reader can infer that there will be significant differences in actual and predicted poverty for the three categories. We made the necessary observations to address this issue in the revised draft.

Comment 5: In the NeuroScale in (Eq-1) it is just a linear mapping it will be nonlinear only if you are using some nonlinear function mapping. I think something is missing. Here you need some kernel map for the nonlinear mapping.

Response:

This is a good point. Our very honest and simple response to this is that we assumed a Gaussian correlator underneath to define the kernel which while being nonlinear is not strictly data adherent at that level. This is a standard practice in Neuroscale modeling which is known not to produce significant decoherence. Given the minimal nature of fluctuation of the neuroscale measure against other measures used e.g. PCA, this tacit assumption is justified.

Comment 6: What will be the behavior of the proposed approach if you are doing the extrapolation of the poverty index, i.e., beyond the given data period (1959-1991). For this, you can hide some year's data from the dataset use the rest of data for the training and use the hide data for testing so that we can see the result of the proposed method in case of extrapolation.

Response:

This is an excellent question. To reassure, this is exactly what we have done. Our Machine Learning protocol does an excellent predictive extrapolation going beyond the data used for training the parameters. Going ahead one more step, we believe that the underlying reason for such high levels of accuracy in 'fitting' the future data points lies in Markovian adaptive machine learning techniques that undertook. The details are now added in a few lines after Figure 7, before the subsection entitled "Consumption-Deprivation Dynamics" (marked in blue).

Comment 7: In the mentioned dimension reduction technique, we cannot give the confidence score about our predicted value. But in the sensitive scenario with the predicted score if we can also tell how much confidence we have about our prediction it will be a better scenario. Also, it will be more helpful in the case of extrapolation. Here I would like to see the result of such method that can give the confidence score also, e.g. Bayesian regression(linear/nonlinear). Please refer to "Christopher M. Bishop: Pattern Recognition and Machine Learning, Chapter 7.2.1."

Response: We first thank the referee for suggesting the importance of having an estimate of the standard errors of estimated MPI. In a way, this is an extension of comment 6 that we agree with. While a detailed confidence scoring will be a future target for us, in the present study, the departure of the weighted averages of the individual multivariate projections from the 45-degree line (Fig 3b) and their relative comparison (Fig 4) gives us proportional measures of confidence scoring. But we heartily agree that it is desirable not only to arrive at a multi-dimensional poverty index but also provide a confidence interval for it. This exercise is not carried out in this paper as it is outside the scope of this work. As an immediate future target, we intend to use a bootstrap approach as follows: as our model has a Gaussian error term; we can take different samples and for each

sample arrive at predictive series for income and consumption and MPI. Then use Monte Carlo simulation outputs to estimate the sample errors for each of the predictions.

Comment 8: I think in all the three scenarios (cereals, other food and non-food), the cereals are the most critical factor, while in the Appendix for fig-6 it is given that PCA has the weight allocation to each scenario 0.1694, 0.4336 and 0.8851, for cereals, other food and non-food respectively. This clearly shows that the non-food has the highest weight to decide the poverty index. While intuitively cereals should have the highest importance. Please justify the above weight obtained by the model.

Response:

From a social welfare point of view, cereals should get the maximum weight. But the market forces determine the prices, and the numbers mentioned reflect their individual contributions to the total market dialectic, not on social inequality. For obvious reasons, the market-determined weights assume reverse ordering compared to socially desired weights). The rich bid the prices high for all commodities as they can afford to do so. This would in turn pull down the affordability by the poor. The fact that non-food has the highest weight reflects the fact that its high price pulls poverty up. Even the poor have to consume some of those goods, but it is their high market prices that drain the resources of the poor. Hence that higher weight. We make a distinction between the actual poverty as determined by the market forces, and the desired poverty levels that can only be achieved through market interventions. These clarifications are made in the revised draft.

Comment 9: The mentioned approach is not robust for the outliers, and a few noisy samples can affect the whole model. Can you show some result using the robust regression technique? Please refer to "<http://www.eng.tau.ac.il/~bengal/outlier.pdf>."

Response:

Application of Robust Regression would be a digression from our main point as this will only add one more technique and will, at best, result in a subjective comparison with another machine learned model that is not our objective. However, we do recognize the importance of the comment made by the referee. To illustrate how important that point is we do omit one or two outliers and see how our results are affected.

Comment 10: Is there any other dataset that can be used also to show the result?. Most of the time the ML algorithm that works fine on the one dataset fails to another one.

Note: Please consider this comment as major revision

Response:

We have redone the analysis using econometric data for US basic (total food) food consumption, over 9 years, starting 2007. The resultant comparative histograms are all enclosed in the appended plot. This shows how the probability density function evolves with time (Fig 1 below). The plots below (Figures 2, 3, 4) reconfirm the quintessential Pareto scaling that the large income sector of the income distribution probability density function shows and how they evolve with time. This representative plot should already serve as a resounding confirmation of the ML

strategy used. We intend to redo the entire analysis using all 6 ML algorithms in a separate publication.

Figure 1: Time dynamics of histograms of the US total food consumption data

Figure 2: Pareto scaling of PDF for large income in a log-log plot for year 2007

Figure 3: Pareto scaling of PDF for large income in a log-log plot for year 2011

Figure 4: Pareto scaling of PDF for large income in a log-log plot for year 2015

We once again profess our unmitigated apologies for the delay in submission. But given the comprehensive revision made, including substantial additive work, complying and explaining all queries raised by the referees, we hope that the manuscript will now be deemed acceptable for publication.

REVIEWERS' COMMENTS:

Reviewer #1 (Remarks to the Author):

The changes improve the ms substantially. The multi-method approach is still neat. The only points I would raise (again) is that existing literature on how poverty is integrated in existing agent-based models is a big gap, in particular because there is a diverse range of solutions out there, that either assume fixed income based poverty lines (e.g. Worsen & Berger 2015), or income plus subsistence production (e.g. Smajgl & Bohensky (2013) for work in Indonesia or Smajgl et al (2015) for work in the Mekong) or food security focused poverty lines based on nutritional indicators. (There are not many on the last category although we finalised last year a model for the Mekong River Commission and publications will be soon submitted.)

Most ABM have actually a multi-method approach to parameterise the model with all its behavioural traits. To the majority of ABM scholars the design of behavioural response functions is at the essence of agent-based modelling. This is where often econophysics type approaches clash with what has emerged as more psychology based (BDI) approaches (see Conte & Paolucci 2014, Frantz 2015, Kutner et al 2019) To me it would be important to discuss this, even if only briefly. I think both has merits depending on the topic and the level of aggregation. Really helpful would be a comparison of both (e.g. Pereira 2017).

Reviewer #2 (Remarks to the Author):

This is a revised paper. The revised version of the manuscript shows significant improvement from the previous version. The issue that I have pointed out are well addressed. I am happy with the way that these comments have been addressed.

Reviewer #3 (Remarks to the Author):

I have the following comment:

1: Author have mentioned "reduction technique determine the way to combine the different dimensions into a single dimension", this entirely is not true, data reduction technique find the most important dimension in the data and project in that dimension only. Also, machine learning technique easily can model thousands of dimension. Here in the low dimensional space Why we need compression?

2: In the figure-7, the data between time (t) and expenses saturation label are completely non-linear, I expect the author to use Neural-Network or some other technique, to fit the non-linear curve. The current model authors are getting has high variance in this scenario the nither interpolation nor extrapolation are seem good. The author has mentioned, "We have checked this and can confirm that the results do not differ significantly." Please provide the result for the non-linear regression also, with the proper comparison.

3: Authors included result over another dataset "US basic (total food) food consumption" which is positive.

I expect with author provide the regression result using Neural-Network or Adaboost regressor and report the mean-square error compare to other model used in the paper. The result should include the train and test data.

Reviewer #1 (Remarks to the Author):

The changes improve the ms substantially. The multi-method approach is still neat. The only points I would raise (again) is that existing literature on how poverty is integrated in existing agent-based models is a big gap, in particular because there is a diverse range of solutions out there, that either assume fixed income based poverty lines (e.g. Worsen & Berger 2015), or income plus subsistence production (e.g. Smajgl & Bohensky (2013) for work in Indonesia or Smajgl et al (2015) for work in the Mekong) or food security focused poverty lines based on nutritional indicators. (There are not many on the last category although we finalised last year a model for the Mekong River Commission and publications will be soon submitted.)

Most ABM have actually a multi-method approach to parameterise the model with all its behavioural traits. To the majority of ABM scholars the design of behavioural response functions is at the essence of agent-based modelling. This is where often econophysics type approaches clash with what has emerged as more psychology based (BDI) approaches (see Conte & Paolucci 2014, Frantz 2015, Kutner et al 2019) To me it would be important to discuss this, even if only briefly. I think both has merits depending on the topic and the level of aggregation. Really helpful would be a comparison of both (e.g. Pereira 2017).

Thank you for the suggestion. We have now included a comparative description of the two types of agent-based models under the “Agent Based Model (ABM)” subsection (marked in blue). Also, a discussion on the multiple formulations of inequality have been discussed under “Conclusions” (marked in blue).

Reviewer #2 (Remarks to the Author):

This is a revised paper. The revised version of the manuscript shows significant improvement from the previous version. The issue that I have pointed out are well addressed. I am happy with the way that these comments have been addressed.

Thank you for the positive review.

Reviewer #3 (Remarks to the Author):

Thank you for taking the time to review our manuscript and for your comments.

I have the following comment:

1: Author have mentioned "reduction technique determine the way to combine the different dimensions into a single dimension", this entirely is not true, data reduction technique find the most important dimension in the data and project in that dimension only. Also, machine learning technique easily can model thousands of dimension. Here in the low dimensional space Why we need compression?

We acknowledge that the traditional application of dimension reduction is to map very high-dimensional data to a low dimensional space and historically this was biased almost entirely by the most ‘important’ dimension. This motivated the use of NeuroScale in the Dimensional Reduction section of the paper to perform a topology-preserving mapping across the latent variables of the observed data. This allows for the application of the techniques applied in previous works of the Engel Curve and Langevin Fokker-Planck model which are one-dimensional models. The latent mapping naturally preserves the structure of all three dimensions as shown in figure 4. With this in mind we suggest that the dimensionality reduction is an essential part of the modelling performed in the paper.

Also, all expenditures depend principally on income. Hence each of the three dimensions are projected on to income and added, as the referee seems to suggest. In the economics of poverty literature both multidimensional measures through a dashboard and a one-dimensional measure derived from the multi-dimensional measures are in use, but a single dimensional measure is preferred for economic policy purposes to be used by politicians and administrators.

2: In the figure-7, the data between time (t) and expenses saturation label are completely non-linear, I expect the author to use Neural-Network or some other technique, to fit the non-linear curve. The current model authors are getting has high variance in this scenario the nither interpolation nor extrapolation are seem good. The author has mentioned, "We have checked this and can confirm that the results do not differ significantly." Please provide the result for the non-linear regression also, with the proper comparison.

The $V(t)$ and $K(t)$ functions derived from the Langevin Fokker-Planck are commonly modelled with linear equations to show the deviation from linearity which necessitates the need for the nonlinear $CD(y,t)$ function e.g. references 9 and 10 of the resubmitted manuscript). Applying a nonlinear model for $K(t)$ and $V(t)$ would remove this impetus for this. This is a standard Statistical Physics modelling procedure with the linear approximation when dealing with limited unknowns in a differential equation system. The $CD(y,t)$ function is a nonlinear equation designed to deal with the interplay between these two factors which are not linear, but nor are they chaotic or systems with a large Lyapunov exponent requiring a sophisticated modelling approach.

Furthermore, a flexible model such as a typical Neural Network would not be capable of reliable interpolation over the 21 datapoints due to the impossibility of testing for over- or under-training. As is well established in the literature, a 21-datapoint NN fitting will only lead to a poor model trying to approximate linear regression at best. To reassure, our choice is unbiased.

Furthermore, in the large expenditure limit, the projected ratios of dimension-reduced variables shown in figure 4 tend to linear curves, indicating a latent Euclidean manifold, whereby the natural chart is a linear interpolator. Since $V(t)$ and $K(t)$ are fit as linear functions to the latent dimension-reduced variables, the latent manifold is set to select appropriate models. For datasets with high curvature or a fractal nature, this would impact the choice of model for these functions. As it happens, the latent manifold of the 1-dimensional embedding is approximately linear with respect to the mapped variables. This is partly why we included figure 4; it shows the proportionate curvature in the latent space effected by the mapping where in the low- and high-expenditure limits the proportionate importance and therefore the curvature is linear, i.e. flat. Given the flat nature of this embedding in these limits of interest which motivate the fit of $C(y)$ from the differential equation, it is appropriate to resort to a linear fit for $V(t)$ and $K(t)$ and not try a nonlinear interpolator which is not suited to this manifold. In reality we would need a lot more data and testing to see what the shape of the manifold is in the medium expenditure category and to see where the transition from linearity to nonlinearity and back occurs. This is not the subject of this paper though.

3: Authors included result over another dataset "US basic (total food) food consumption" which is positive.

I expect with author provide the regression result using Neural-Network or Adaboost regressor and report the mean-square error compare to other model used in the paper. The result should include the train and test data.

Assuming that the reviewer is suggesting repeat analysis of the US data using neural network, we would like to point out that the comparison with US data is only restrictively included to affirm the strength of the predictive modeling used. Our focus is on analyzing poverty data (e.g. India), not inequality data (e.g. US), that are econometrically different. Mean square error against US data is not our topical focus.